# Short tandem repeat variants are possibly associated with RNA secondary structure and gene expression

**Nick Kinney** **¤\*, Dikshya Pathak, Emma Evans, Paola Arias**

Sweet Briar College, Sweet Briar, Virginia, United States of America

¤ Present Address: 134 Chapel Road, Sweet Briar, VA 24595, United States
\* nkinney@sbc.edu

## Abstract

Short tandem repeats (STRs) are abundant in the human genome with approximately 300,000 embedded in gene introns, exons, and untranslated regions. High penetrance STR variants cause human diseases such as Myotonic dystrophy, Baratela-Scott syndrome, and various ataxias. The possibility that STRs contribute to polygenic disease is supported by recent high-powered datasets that link STRs to more subtle effects on gene expression. Indeed, STR variants can induce Z-DNA and H-DNA folding; alter nucleosome positioning; and change the spacing of DNA binding sites. On the other hand, little is known about how STR variants affect RNA secondary structure and accessibility. These factors could affect rates of splicing, nuclear export, and translation. We hypothesize that effects on RNA structure can be predicted using computational tools and associated with gene expression using DNA and RNA sequencing data. We test this hypothesis using data from the 1000 Genomes Project and ViennaRNA. We identify 17,255 transcribed STRs that affect RNA folding (fSTRs); 356 are possibly associated with gene expression. We characterize fSTRs by repeat motif, length, and gene level annotation. Transcribed fSTR variants tend to affect RNA multiloops and external loops. Effects on RNA accessibility depends on the repeat motif: a surprising result that is checked against simulation. These results shed light on how transcribed STRs affect RNA structure and pave the way for experimental validation.

## Introduction

Short tandem repeats (STRs) are hotspots for human genetic variation [1]. Their repetitive sequence motifs (1–6 base pair) are prone to strand slip replication and unequal crossing over which tend to increase or decrease the STR array length [1,2]. Indeed, STRs have been used for decades as markers in forensic and population analysis [3,4]. Approximately 300,000 STRs are embedded in human gene introns,

**Data availability statement:** We used freely available python packages to perform our analysis. In addition to those already mentioned, we use sklearn to preform affinity propagation clustering on RNA structural similarity scores. We use the force directed RNA (forna) web interface to produce secondary structure plots of select STRs (71): http://rna.tbi.univie.ac.at/forna. All other plots were prepared with plotnine and pillow for python. Data and code used for manuscript preparation are freely available as supplementary material and online: https://github.com/nkinney06/.

**Funding:** The author(s) received no specific funding for this work.

**Competing interests:** The authors have declared that no competing interests exist.

exons, and untranslated regions (UTRs); consequently, variation in these regions is possibly associated with differential gene expression across human populations [5,6]. In fact, this hypothesis has recently been supported and reproduced by integrating data from DNA and RNA sequencing [7,8].

In 2015 and 2019, a pair of studies used variance partitioning to survey the human genome for STRs associated with gene expression [7,8]. The first study identified 2,060 expression STRs (eSTRs). The second study identified 28,375 eSTRs and recapitulated many of the 2,060 identified in 2015 [7]. The discovery of correlations between eSTR array length and gene expression provides a measure of validation for past and future studies of STRs in complex disease. In fact, several studies prior to 2015 reported links between various cancers and STR variation [9,10]. Since then, STR variation has been investigated in several additional cancer types [11,12] and autism spectrum disorder [13]. These breakthroughs paved the way for dedicated catalogues of STR variation [14–16]. In particular, WebSTR provides a catalogue of genome-wide STR variation in humans, and currently contains data for approximately 1.7 million unique regions [14].

The idea that STRs can affect gene expression is not surprising. Specific STR variants are causative in various ataxias, Huntington's disease, and fragile X syndrome [17]. These high impact examples have been known for decades; however, discovery of more subtle effects on gene expression have had to wait for large datasets with more statistical power. These data have helped link STR variation to complex traits including blood and lipid biomarkers as well as oxidative stress [5,18]; and, the aforementioned studies of cancer and autism. The results suggest the possibility that STR variations can be leveraged for diagnostic proposes [19]. This hypothesis is supported by several studies of human cancer; in particular, colorectal and breast cancer [20,21]. So far, most of the attempts to leverage STRs for diagnostic purposes have used a polygenic risk approach with modest results [11,12,21].

The mechanisms that dictate how STR variants affect gene expression are diverse with some known and some unknown. Regardless of their position in the genome, STR variants can inducing Z-DNA and H-DNA folding [22]; alter nucleosome positioning [22,23]; and change the spacing of DNA binding sites [22,24]. When STR variants are positioned in coding regions they have the additional capacity to affect protein folding. Due to the possibility of frameshift, STRs embedded in coding regions are under unique selective pressure that favors insertion and deletion (indel) factors of three [25–27]. In addition, those coding for hydrophilic amino acids are overrepresented [27]. Indeed, polyglutamine variants are among the most common of the repeat expansion disorders [28,29].

Relatively few studies have investigated how transcribed STR variants affect RNA structure [30–32]. This is important because a precedent has been set that links RNA structure to gene expression in humans. RNA sequence (primary structure) can affect translational speed and accuracy when the transcript's 5' end is enriched with rare, slowly-translated codons [33–35]. The folding of RNA into hairpins, loops, and other structural motifs (secondary structure) can affect how the RNA interacts with proteins, ribosomes, and other RNAs [36,37]. However, links between STR variants and

possible effects on RNA secondary structure are understudied. We hypothesize that some transcribed STRs affect RNA secondary structure which in turn are associated with gene expression. If supported, this hypothesis would contribute to what is known about how STRs affect differential gene expression across populations and disease states.

We use a data integration approach to test our hypothesis. Briefly, STR variants are identified from samples in the 1000 genomes project [38]. We focus on transcribed variants found in intron, exon, UTR, and coding regions. Next, we use the ViennaRNA package to predict the secondary structure of each variant [39]. To identify STRs that affect RNA folding (fSTRs) we cluster each collection of secondary structures using bpRNA-align [40,41]. Briefly, bpRNA-align uses a state-of-the-art global structural alignment algorithm to improve clustering performance over a broad range of structure types [41]. Finally, fSTRs are tested for association with gene expression using 462 human lymphoblastoid cell line samples created by the Geuvadis consortium [42]. We characterize fSTRs by motif length, gene level annotation, and effects on RNA folding. We discuss our results in the context of recent STR studies and suggest future lines of inquiry.

## Results

### Transcribed STRs are possibly associated with gene expression

The overall goals of this study are threefold: (a) identify STR variants that affect RNA folding (fSTRs); (b) establish an association between fSTRs and gene expression; and (c) characterize the effects of fSTR variants on RNA folding. To begin we identify STR variants in 2,529 samples from the 1000 genomes project [38]. Variants for each transcribed sequence – including 50 bp of 3' and 5' sequence – were folded with ViennaRNA [39]. Secondary structures were compared with bpRNAalign and affinity propagation clustering [40,41]. Changes in RNA structure were indicated by clustering results in excess of one (Fig 1, left panel). We identify 17,255 fSTRs. A representative fSTR in an intron of SH2B3 has five variants with transcribed RNA sequences forming two clusters (Fig 1, left panel).

Associations with gene expression were checked using RNAseq data for a subset of the samples. We use the 462 human lymphoblastoid cell line samples created by the Geuvadis consortium [42]. The analysis was performed in three steps. First, variants from those samples were mapped to their corresponding cluster assignments. Second, expression for genes harboring fSTRs were grouped by genotype; i.e., a pair of variants mapped to cluster assignments (Fig 1, right

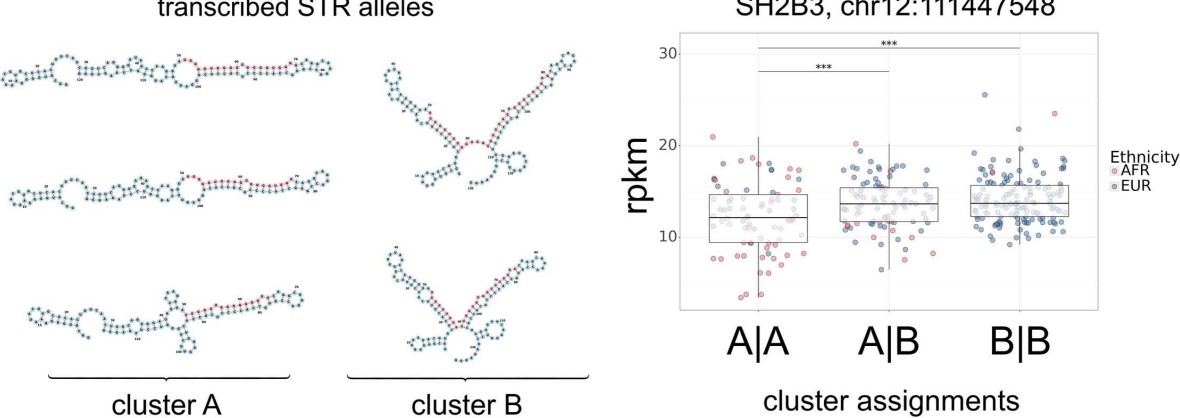

**Fig 1. Short tandem repeat (STR) variants affect RNA secondary structure and are possibly associated with gene expression.** (left) The effect of STR variants on RNA folding (fSTRs) was inferred by comparing secondary structures with bpRNA-align and affinity propagation clustering. Five variants of a penta-repeat (TGGGG) in an SH2B3 intron fall into two clusters (A and B). (right) Gene expression (rpkm) values for a collection of samples are grouped by genotype after mapping each variant to its cluster assignment. Since each sample has two alleles there are three combinations of cluster assignments (independent axis). We perform a test of the null: no difference in gene expression between groups. The null is rejected (p < .01) suggesting an association between RNA folding and SH2B3 expression.

panel). Third, we perform a test of the null: no difference in gene expression between groups. The null is rejected for 356 fSTRs suggesting an association with gene expression. Cluster assignments for an fSTR in SH2B3 show significant differences in gene expression (Fig 1, right panel).

## fSTRs are over represented in coding regions

We reiterate the discovery criteria for a single fSTR: affinity propagation of its transcribed variants forms two or more clusters. The 66,876 transcribed STRs investigated in this study revealed 17,255 (25.8%) fSTRs. However, this may be an underestimate for two reasons. First, we only considered variants identified in the 1000 Genomes Project. Additional STR variants would likely be found with a larger sample size. Undoubtedly, some of the single cluster results would form multiple clusters with these additional variants. Second, STRs lacking variation in the 1000 Genomes Project samples were excluded from analysis: without variation there is no suitable test of the null. A larger set of samples would likely reveal variation in some of the excluded STRs and the discovery of additional fSTRs.

Characterization of fSTRs by gene level annotation reveals overrepresentation in coding regions (Fig 2a). This result is intriguing when paired with characterization of fSTRs by motif length. It comes as no surprise that effects on RNA structure increase with motif length; indeed, motif lengths greater than one are overrepresented (Fig 2b). However, coding regions are known to favor motif lengths of 3 and 6 to avoid frameshift. Apparently, coding regions are under far greater selective pressure to avoid frameshifts than fSTRs. If this were not the case, the unit one motifs – underrepresented among fSTRs – would outnumber unit three motifs in coding regions.

Characterization of fSTRs by unit is harder to interpret (Fig 2c). The well-known CAG motif (listed as its equivalent ACG motif) is conspicuously associated with fSTRs, but so too are many other motifs. Taken as a negative result, one interpretation is that any motif has the capacity to affect RNA folding.

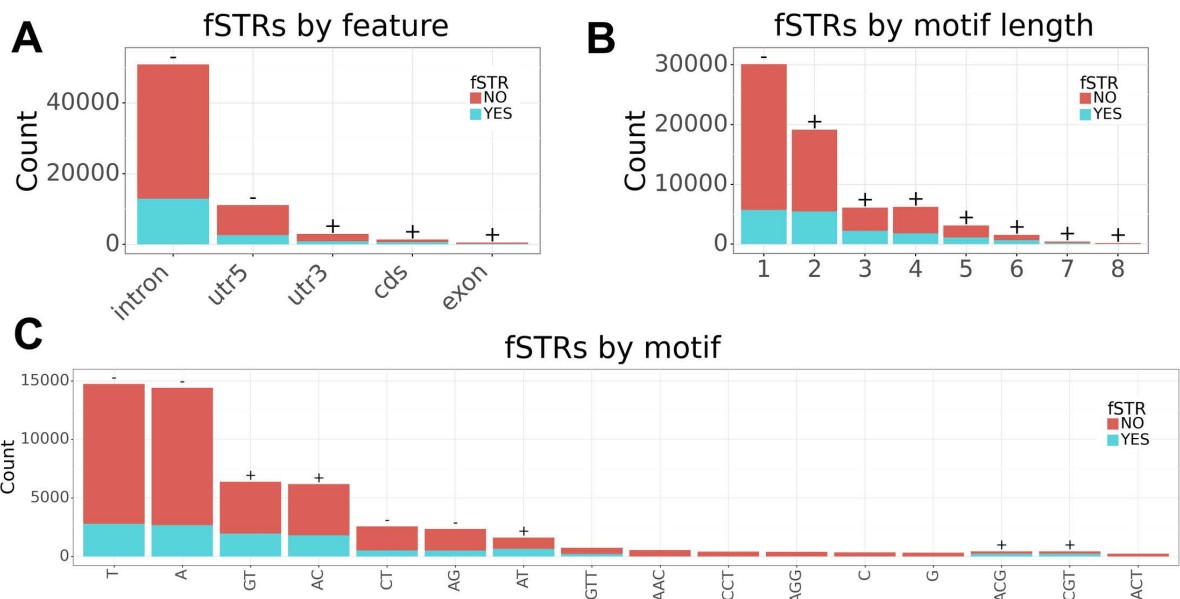

**Fig 2. Characterization of fSTRs by gene level annotation, unit length, and repeat motif.** (a) fSTRs are overrepresented in coding regions. (b) effects on RNA structure tend to increase with motif length. (c) characterization of fSTRs by sequence motif. (+) over-representation among all fSTRs. (-) under-representation among all fSTRs.

## fSTR motifs affect RNA accessibility

RNA accessibility may be important for protein binding, rates of splicing, nuclear export, and translation. We characterize how fSTRs affect accessibility of minimum free energy (MFE) RNA structures using ViennaRNA. Briefly, the core prediction algorithm uses dynamic programming to predict base paired and unpaired regions within single stranded RNA. To infer accessibility, we tally unpaired bases for fSTRs and stratify the results by allele length and repeat motif (Fig 3). Results of two types are obtained: (a) accessibility increases with allele length; (b) accessibility decreases with allele length. Examples of both types are shown in Fig 3a and 3b, respectively. Although strong examples were found for both types of association; accessibility varies substantially for fSTR alleles of fixed length regardless of motif. Thus, RNA accessibility probably depends on the fSTR allele length as well as the sequence context 5' and 3' to the actual repeat motif.

To further characterize RNA accessibility, we investigate possible associations with repeat length and unit. Associations of this type are hard to pin down with one exception. Sequences serving as their own reverse complement tend to decrease accessibility as allele length increases. For example, the reverse of poly-AT (poly-TA) is complementary to the original poly-AT motif (Fig 3b). We speculate that such sequences – which have the ability to base pair with themselves – cause a decrease in transcribed RNA accessibility. To test this, we aggregated all non-reverse complementary and reverse complementary sequence motifs. Indeed, the former sequence motifs show a positive correlation with allele length (Fig 3c; $r = 0.017$, $p = 1.3e\text{-}10$) while the latter have a negative correlation (Fig 3d; $r = -0.214$, $p = 0$).

## fSTRs tend to affect RNA multiloops and external loops

The effects of fSTRs on MFE RNA folding are characterized by comparing secondary structure motifs using bpRNA and bpRNA-align [40,41]. Briefly, the per base secondary structure assignments are aligned for each pair of variants

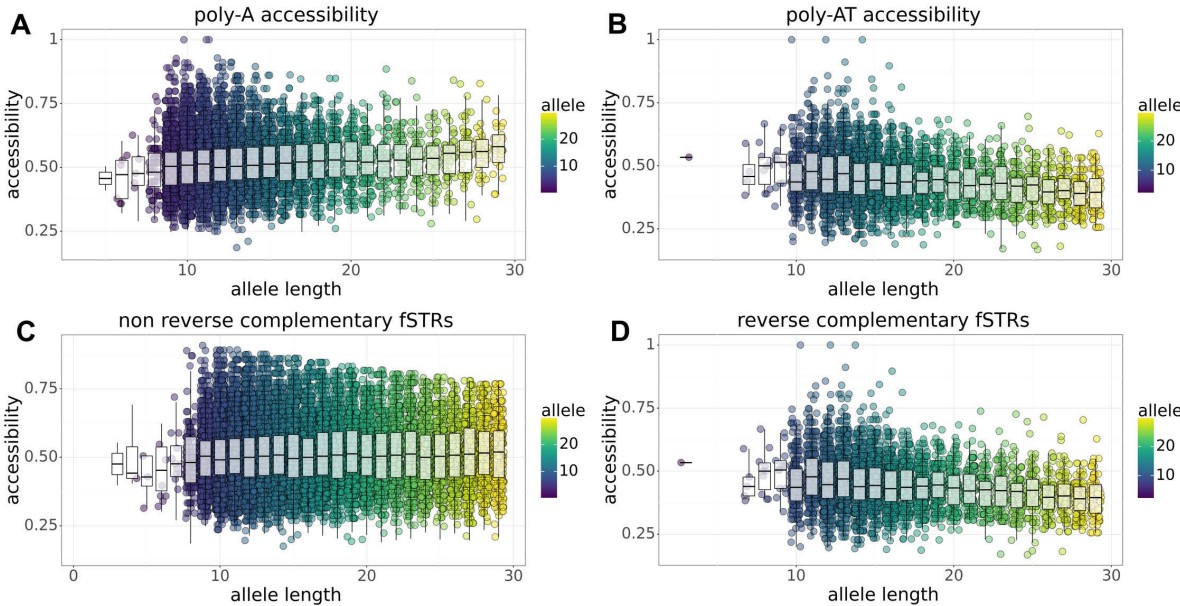

**Fig 3. Effects of fSTR variants on RNA accessibility for MFE secondary structures.** Accessibility is inferred from the tally of unpaired bases using ViennaRNA. (a) Accessibility increases with allele length for poly-A repeats: $r = 0.066$, $p = 0$. (b) Accessibility decreases with allele length for poly-AT repeats: $r = -0.217$, $p = 0$. (c) Accessibility increases with allele length for non-reverse complementary repeats: $r = 0.017$, $p = 1.3e\text{-}10$. (d) Accessibility decreases with allele length for reverse complementary sequences: $r = -0.214$, $p = 0$.

belonging to an fSTR. Mismatching structural motifs are tallied over pairs of alleles. Tallies are visualized as a matrix with row sums normalized to 100% and columns indicating the frequency of mismatch with all other motifs. Over 15% of RNA multiloops (M) and external loops (X) are affected (Fig 4a); and, they are frequently exchanged with one-another. Frequent changes to bulge motifs (B) are also common (red off-diagonal in Fig 4a). Interestingly, no changes are prohibited. Dangling end motifs (E) were rarely exchanged for multiloops (M) with the former being altered in less than 3% of the bases tallied (Fig 4a).

For reverse complementary sequences (see previous section) we notice a many to one shift towards left (L) and right-handed (R) stem motifs: these columns are mostly red for reverse complementary motifs (Fig 4f). We see a shift away from multiloops (M) and external loops (X) suggesting a link between some fSTRs and gene expression. Indeed, multibranch loops (M) are hubs of interaction within RNA. In fact, this is precisely the difference seen between the clusters of variants for the fSTR embedded in SH2B3 (Fig 1a). However, that particular repeat is not reverse complementary. Of course, the suggested links between DNA motif, RNA folding, and gene expression should be interpreted as preliminary associations and not causation.

## Simulations recapitulate effects of STR variants on RNA structure

The effects of reverse complementary sequences were verified using a simulation-based approach. This is important for two reasons. First, sequences 5' and 3' to repeat variants may influence RNA folding as seen in experiment. Second,

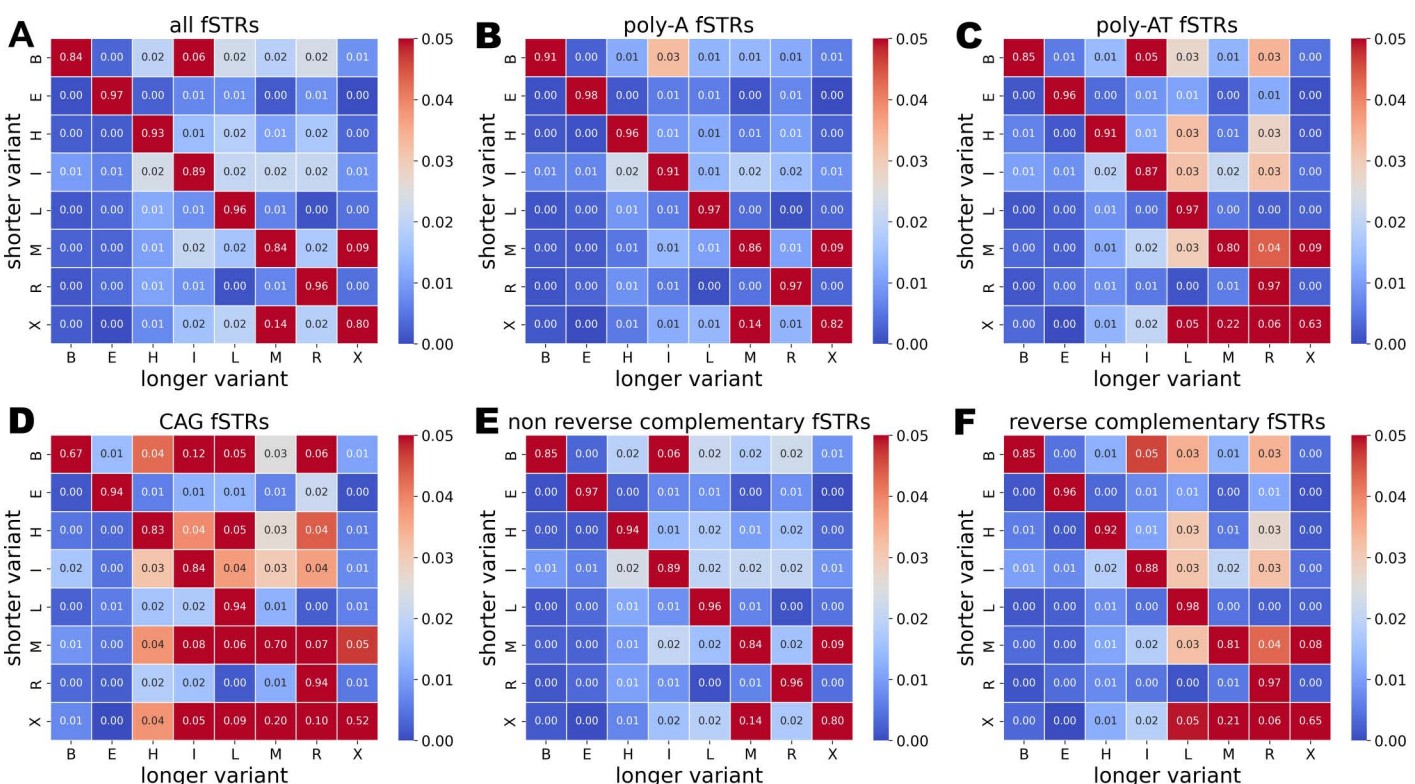

**Fig 4. MFE secondary structure changes tallied for pairs of fSTR alleles and normalized by row: left-handed stem (L), right-handed stem (R), internal loops (I), bulges (B), hairpins (H), multiloops (M), external loops (X), and ends (E).** (a,b) Multiloops and external loops are frequently exchanged due to fSTR insertions. (e) The same exchange is seen for non-reverse complementary sequences. (c) The AT motif – which base pairs with itself – shifts towards right (R) and left-handed stem (L) motifs. (f) The same shift is seen for reverse complementary fSTRs. (d) CAG repeats conserve right-handed stems (R), left-handed stems (L), and ends (E) while departing from other structural motifs.

singleton motifs are overrepresented in the experimental data. Accessibility was tested on 10,000 simulated STR alleles. In each case, 5' and 3' sequence context was randomized. Reverse complementary (Fig 5a) and non-reverse complementary (Fig 5c) motifs were sampled randomly. The results recapitulate the experimental data in Fig 3c and 3d, respectively.

Effects on secondary structure used a similar approach. Motifs were sampled randomly. Changes in RNA secondary structure were tallied for five simulated indel variants while keeping the 5' and 3' sequence context fixed. Simulations for reverse complementary sequences (Fig 5d) recapitulate experimental data (Fig 4f). However, the remaining sequences (Fig 5b) do not recapitulate experimental data (Fig 4e). The difference undoubtedly stems from the aforementioned over-representation of singleton motifs in experiment. Interestingly, the different motifs have little effect on secondary structure in simulation (Fig 5b and 5d). Apparently, reverse complementary sequences affect RNA accessibility (but not structure) while singleton motifs affect RNA secondary structure (but not accessibility).

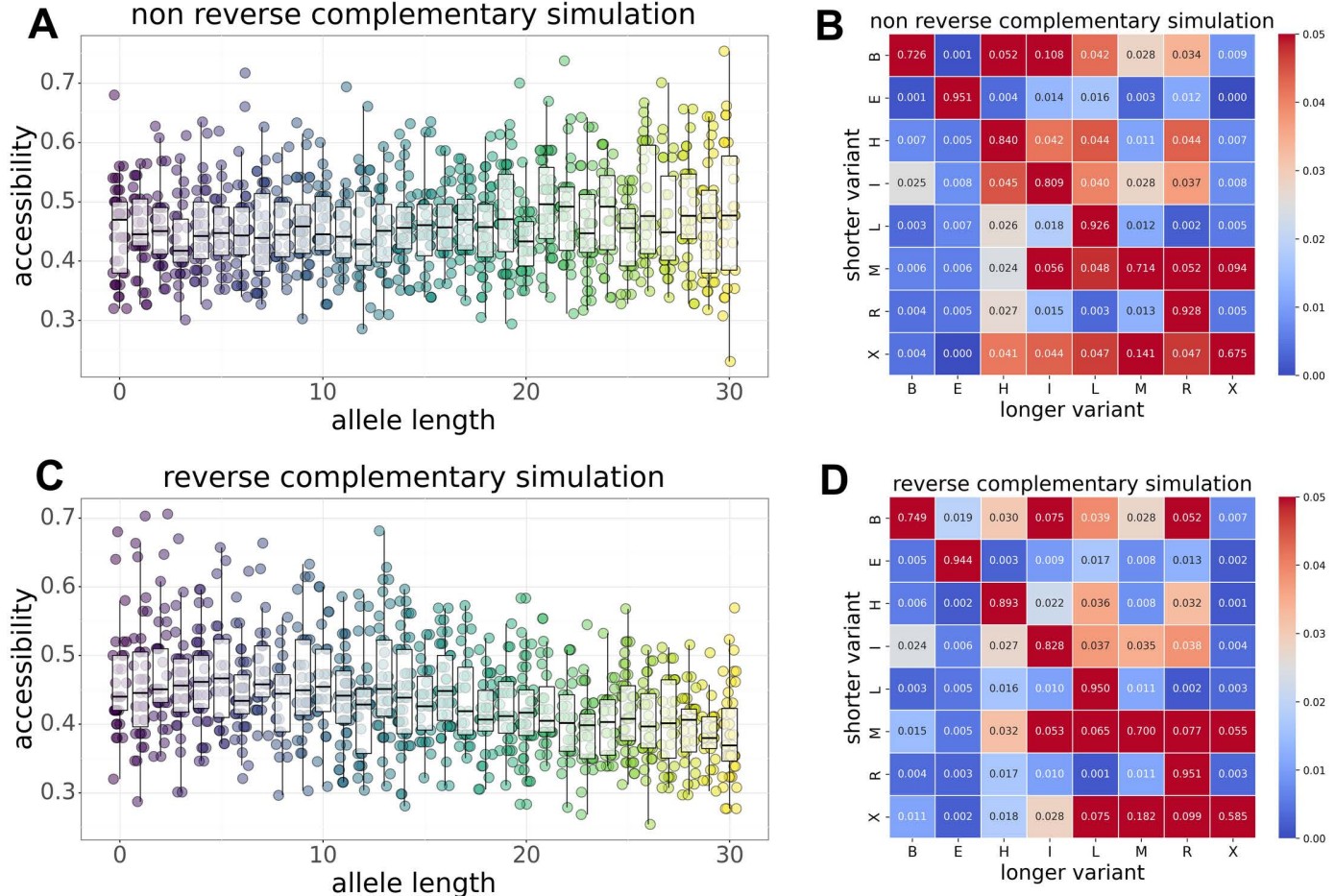

**Fig 5. The effects of fSTR variants on MFE RNA structures was verified using a simulation-based approach.** (a) accessibility increases with motif length for non-reverse complementary sequences. (b) changes in secondary structure for non-reverse complementary sequences. (c) accessibility decreases with motif length for reverse complementary sequences. (d) changes in secondary structure for reverse complementary sequences.

## Discussion

Our results support the hypothesis that some STR variants affect RNA secondary structure and gene expression. Support is provided by several lines of evidence. First, we fold and cluster variants for 66,876 transcribed STRs from the 1000 genomes project using ViennaRNA and bpRNA-align. We find 17,255 affect RNA folding (fSTRs). Interestingly, fSTRs are enriched in coding regions and specific 3-mers which conspicuously include the CAG repeat motif (Fig 2c). Although the collection of 17,255 fSTRs are discovered using computational tools, we emphasize that only real variants identified in 1000 Genomes Project samples were used for the analysis. Next, we infer effects on gene expression using RNAseq. Briefly, we map fSTR clusters to RPKM values for each sample and preform a test of the null: no association between cluster assignment and RPKM. Association is supported for 356 fSTRs. These include 13 in coding regions: SAAL1, ZNF384, TSC22D1, MEF2A, C16orf71, TOX3, ERN1, NADK, PTPN18, GIGYF2, USF3, TRERF1, and AK9.

Not to be lost in our results is the approach itself. We demonstrate a novel way to study STR variation using state of the art tools. ViennaRNA is widely regarded as the best in class for predicting RNA secondary structure and bpRNA-align is a recent addition that shows improvement in clustering performance over a broad range of structure types [39,41]. This approach could be extended to study other classes of repetitive DNA such as palindromes and terminal inverted repeats. Indeed, similar approaches have been used – with an older set of tools – to study the effects of single nucleotide polymorphisms (SNPs) on the structure of transcribed UTRs and RNA in general [43–45]. Most of the novelty we introduce lies in mapping the bpRNA-align cluster assignments to variants possessed by each sample; a critical step that enables RPKM association testing.

Our approach is easily extended to the study of disease provided both DNA and RNA sequencing data is available. This is certainly the case for many samples in The Cancer Genome Atlas (TCGA) and database of genomes and phenomes (dbGaP). However, the idea that RNA folding alone is sufficient to explain high impact STR variants should be approached with skepticism. Those that are known have catastrophic effect on protein structure (such as Huntington's) or chromosome structure (such as fragile X); but not RNA structure. In other cases, epigenetic modifications (such as CpG methylation) may overshadow the effects of array length polymorphisms by silencing genes prior to transcription altogether. It is more reasonable to conclude that RNA structure alterations have modest effects on rates of transcription, translation, and splicing.

Beyond splicing, RNA secondary structure influences post-transcriptional gene regulation, particularly when variants occur in untranslated regions (UTRs) or coding sequences [31,46]. Variants in the 5′ UTR may modulate translation initiation while those in coding may affect elongation rates. Variants in the 3′ UTR may impact transcript stability or localization by disrupting motifs for RNA-binding proteins. Future work integrating ribosome profiling, RNA stability assays, and RNA binding protein mapping will help clarify and validate the broader functional consequences of fSTRs.

On the contrary, STR variation and its influence on RNA structure could play a larger role in prokaryotes where transcription and translation are spatially and temporally linked. In fact, two processes unique to prokaryotes provide a precedent. Attenuation is a well-established mechanism that leverages codon repeats to regulate transcription via mutually exclusive RNA secondary structures [47,48]. Possibly any STR variation that alters RNA secondary structure could influence the rate transcription or lead to its termination all together. While this is just a hypothesis, it may be experimentally tractable. A second process – bacterial phase variation – leverages STR mutation rates for semi-random dichotomous phenotype variation [49,50]. Although phase variation has more to do with DNA structure than RNA structure, it emphasizes the complex role of STR variation on phenotype.

To validate our computational predictions regarding RNA-protein interactions and translation efficiency, several experimental techniques could be employed. Cross-linking immunoprecipitation (CLIP) methods, such as HITS-CLIP or iCLIP, allow for transcriptome-wide mapping of protein binding sites on RNA at nucleotide resolution [51]. Applying CLIP to our system would test whether predicted RNA variants alter protein binding in vivo. Similarly, SHAPE-seq and DMS-seq could

provide experimental insight into RNA secondary structure changes caused by fSTR variants [52,53]. For translation efficiency, ribosome profiling (Ribo-seq) offers a powerful means to assess ribosome occupancy along transcripts [54]. Comparing ribosome footprint density across transcript variants could determine if fSTR variants influence translation in vivo. When used in parallel with RNA-seq from the same samples, Ribo-seq also enables calculation of translational efficiency ratios, providing a direct test of our predictions. Together, these approaches offer complementary validation strategies that could substantiate the functional effects of fSTRs proposed in this study.

We suggest further lines of inquiry to investigate the effects of STR variation on RNA and DNA structure. The secondary structure of DNA may affect rates of transcription and protein interactions: both precursors to gene expression. Prediction of Z-DNA, H-DNA, and cruciform DNA are obvious starting points; but, newer tools offer a more sophisticated approach to DNA structure prediction. Deep DNAshape predicts up to a dozen intra-base and inter-base features which could shed light on how STR variation affects transcription factor binding and DNA-protein binding at large [55]. RhoFold uses a language model based deep-learning approach to predict the 3D structure of RNA which could extend our analysis of secondary structure to tertiary structure [56]. Likewise, tools for predicting ramp sequences could provide a starting point for linking STR variation to translation rate and fidelity [34,35].

## Methods

### Overall approach

Our overall hypothesis is that some STRs affect RNA folding (fSTRs) which in turn is associated with differential gene expression in human populations. A test of our hypothesis unfolds in two parts. First, we identify which (if any) of the 66,876 transcribed STRs in the human genome have the capacity to affect RNA folding (secondary structure). To do this, we use the ViennaRNA package to predict secondary structures and score their differences with bpRNA-align. We find 17,255 fSTRs which we characterize by repeat length, repeat motif, and functional annotation. Details of RNA folding and clustering are provided below. Next, we identify which (if any) of the fSTRs are possibly associated with gene expression (Fig 6).

To check for association with gene expression We use a second set of 462 RNAseq samples. Alleles for each sample were mapped to their transcribed cluster assignments (see below). Differences in gene expression (measured as RPKM) across cluster assignments were assessed with a post-hoc Tukey's Honestly Significant Difference (HSD) test. The test

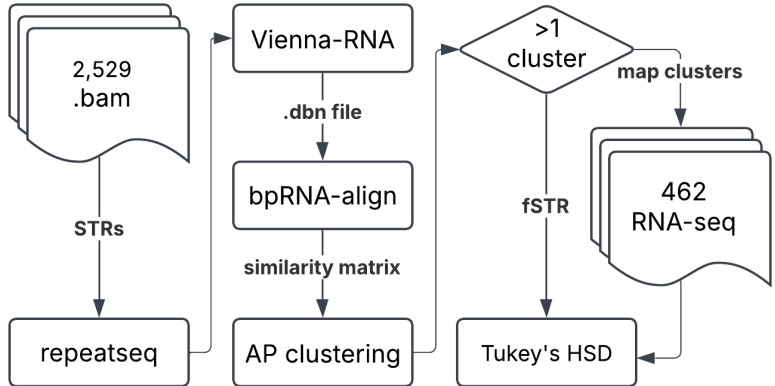

**Fig 6. Summary of overall approach.** (left) Variants for 66,876 transcribed STRs were identified in 2,529 samples from the 1000 Genomes Project. We used repeatseq: a standard STR variant caller. (middle) Variants for each STR were transcribed and folded using Vienna-RNA. Secondary structures were assigned and clustered with bpRNA-align and affinity propagation clustering, respectively. (right) Effects on RNA folding were indicated by clustering results in excess of one. Cluster assignments were mapped to 462 RNA-seq samples: a subset of the original 2,529 samples. Associations with gene expression was established using a Tukey's Honestly Significant Difference test.

was conducted using the pairwise_tukeyhsd function in Python, with RPKM values as the dependent variable (endog) and group assignments based on allele clusters as the independent variable (groups). A significance threshold of $\alpha = 0.05$ was applied to determine pairwise differences between groups. This approach allowed for the identification of statistically significant differences while controlling for multiple comparisons: 356 fSTRs were possibly associated with gene expression (Fig 6). Data and code used for manuscript preparation are freely available online: https://github.com/nkinney06/fSTRs.

## RNA folding with ViennaRNA

The ViennaRNA package is a widely used software suite for predicting and analyzing RNA secondary structures [39]. Briefly, it employs thermodynamic models to predict the most probable secondary structure of an RNA sequence. The core prediction algorithm uses dynamic programming to find the minimum free energy (MFE) structure, which is considered the most stable structure according to the energy model. Details of the ViennaRNA algorithm can be found elsewhere.

Input to the package typically consists of a single RNA sequence or a set of aligned sequences. For single sequences, the RNAfold program can predict either the MFE structure or thermodynamic ensembles using the partition function approach.

In our case, STR variants are inferred from 1000 Genomes Project samples using Repeatseq [57]: http://github.com/adaptivegenome/repeatseq. Details of variant calling are provided below. Each variant is transcribed and saved in fasta format to serve as input to ViennaRNA. ViennaRNA provides dot bracket notation (.dbn) output for each variant. A list of dbn files serves as the starting point for bpRNA-align clustering.

## Thermodynamic considerations

The use of MFE structures without taking into consideration thermodynamic ensembles for each variant may raise concerns about our methodology. In reality, each variant folds into an ensemble of structures approximately $1k_bT$ around the MFE structures. It's conceivable that the energy barrier between some MFE structures is less than $1k_bT$; consequently, the similar overlapping ensembles mitigate any biological effects. This possibility may increase the false positive rate for the 17,255 fSTRs; but not the 356 fSTRs possibly associated with gene expression. Indeed, strong associations with gene expression are inconsistent with weak energy barriers between ensembles.

## RNA clustering

We use bpRNA-align to compare structural differences between STR variants [40,41]. Details of bpRNA-align can be found elsewhere. Briefly, it is a recent contribution that uses a customized global (Needleman-Wunsch) dynamic programming approach. Per base mismatches are scored with a feature-specific substitution matrix and coupled with an inverted and context-specific affine gap penalty. The approach shows improvement in clustering performance over a broad range of structure types [41]. In our case, a list of dbn files (from ViennaRNA) serves as the starting point for bpRNA-align clustering. The output is a symmetric matrix of pairwise similarity scores for each variant. We use the matrix of similarity scores to cluster RNA secondary structures.

Clustering was performed using affinity propagation [58]: the same approach used by the authors of bpRNA-align [41]. We use the AffinityPropagation function from sklearn with the precomputed bpRNA-align similarity matrix. Changes in RNA structure were indicated by clustering results in excess of one. We use a filter parameter to mitigate false discoveries. Briefly, entries in the bpRNA-align similarity matrix were compared for each cluster. Only clusters with differences in excess of 100 were considered for analysis.

## RNA Sequencing (RNA-seq) Data Analysis

The RNA-seq data analysis unfolded in four steps [59]: (a) quality control and preprocessing, (b) alignment to the human reference genome, (c) read counting, and (d) differential expression analysis.

(a) **Quality Control and Preprocessing.** Quality assessment of the sequencing reads was performed using FastQC [60]. Commonly expected warnings, such as sequence duplication due to highly expressed transcripts and minor issues with tile quality, were disregarded. Similarly, K-mer content warnings arising from random priming were ignored, as our analysis focused on gene-level counts rather than alternative splicing or de novo gene structure inference [61].

(b) **Alignment to the Human Reference Genome.** Reads were aligned to the GRCh38 human reference genome using STAR (Spliced Transcripts Alignment to a Reference) [62]. This tool is optimized for handling reads with insertions and deletions. The alignment utilized GENCODE annotation release 33 (gencode.v33.annotation.gtf) to enhance accuracy.

(c) **Read Counting.** Gene-level read counts for each sample were generated using HTSeq [63]. Exon-level counts (--type＝exon) were aggregated by gene ID (--idattr＝gene_id) without strand specificity (--stranded＝no). Counts were subsequently normalized to FPKM (fragments per kilobase of transcript per million mapped reads) using the countToFPKM package in R.

(d) **Differential Expression Analysis.** DESeq2 [64] was employed to identify differentially expressed genes between 89 African and 373 European samples. The analysis began with constructing a count matrix where rows represented genes and columns corresponded to individual samples. DESeq2 automatically estimated size factors, computed gene-level dispersion, and fitted a generalized linear model to identify significant differences.

## STR genotyping Using Repeatseq

Microsatellite genotypes were inferred from whole-genome sequencing data using RepeatSeq, a Bayesian framework specifically designed for genotyping tandem repeats from short-read sequencing datasets. RepeatSeq models PCR stutter noise, sequencing errors, and allele sampling to probabilistically call the most likely genotype at each locus. Input data consisted of aligned BAM files from the 1000 Genomes Project, which were processed according to the developers' recommendations. Candidate repeat loci were specified in BED format, and reads overlapping these regions were extracted for analysis. For each locus and sample, RepeatSeq calculates genotype likelihoods by comparing observed read counts of repeat lengths to a stutter noise model fitted during analysis. The program reports maximum likelihood genotype calls as well as posterior probabilities, allowing for quality filtering in downstream analyses. Default parameters were used unless otherwise specified, with a minimum read coverage threshold applied to ensure reliability of calls. RepeatSeq has been used in previous studies and is freely available online: https://github.com/adaptivegenome/repeatseq. Additional details of STR genotyping are provided in our previous publications [5,6].

When benchmarked on diverse datasets, several recent variant callers report similar or better accuracy than repeatseq such as GangSTR [65], HipSTR [66], lobSTR [67], STRetch [68], TREDPARSE [69], and Dante [70]. Our use of RepeatSeq was justified in our previous publication. In particular, RepeatSeq was specifically designed and validated using data from the 1000 Genomes Project [57].

## Samples

Samples used to identify fSTRs can be found in previous publications. Briefly, these samples come from phase 3 of the 1000 Genomes Project: ftp://ftp.1000genomes.ebi.ac.uk/vol1/ftp/phase3/. In total, 2,529 samples were included for analysis: 667 African (AFR), 502 European (EUR), 352 American (AMR), 514 East Asian (EAS), 494 South Asian (SAS). We use a second set of 462 RNAseq samples for association testing of fSTR cluster assignments against RPKM values. These include 89 Africans and 373 Europeans. All samples are available through the European Bioinformatics Institute website: https://www.ebi.ac.uk/arrayexpress/experiments/E-GEUV-1/samples/.

## Statistical considerations

To evaluate pairwise differences between binned datapoints, we used Tukey's Honestly Significant Difference (HSD) test, which is specifically designed for post-hoc comparisons following ANOVA. This method controls the family-wise error rate (FWER), reducing the likelihood of false positives that can arise from multiple testing. Tukey's HSD achieves this by adjusting the significance threshold across all pairwise comparisons, ensuring that the overall probability of making one or more Type I errors remains at the specified alpha level (typically 0.05). As such, it provides a conservative and statistically robust approach to identify significant group differences while accounting for the multiple comparisons inherent in our analysis.

While it is true that multiple testing corrections can be applied both within and across families of tests, we chose to apply Tukey's Honestly Significant Difference (HSD) test within each binned comparison group without an additional layer of correction across bins. This decision reflects our aim to identify localized effects of specific variants or sequence contexts, rather than to make broad claims about global significance across the entire dataset. Tukey's HSD already controls the family-wise error rate for the multiple pairwise comparisons within each group, which are the relevant statistical units for our hypotheses. Furthermore, because each bin represents a biologically distinct context, we treat these as independent analytical units rather than as components of a single multiple testing framework. As such, we interpret statistical significance conservatively and contextualize findings based on consistency across bins and biological plausibility, rather than relying solely on adjusted p-values for global inference.

## Supporting information

**S1 File. Expanded characterization of STRs, fSTRs, and efSTRs.** Each is characterized by gene feature, sequence motif, and amino acid motif.
(PDF)

## Author contributions

**Conceptualization:** Nick Kinney, Emma Evans, Paola Arias.

**Data curation:** Nick Kinney.

**Formal analysis:** Nick Kinney, Dikshya Pathak, Emma Evans, Paola Arias.

**Funding acquisition:** Nick Kinney.

**Investigation:** Nick Kinney, Dikshya Pathak, Emma Evans, Paola Arias.

**Methodology:** Nick Kinney, Dikshya Pathak, Emma Evans, Paola Arias.

**Project administration:** Nick Kinney.

**Resources:** Nick Kinney.

**Software:** Nick Kinney, Dikshya Pathak, Emma Evans, Paola Arias.

**Supervision:** Nick Kinney.

**Validation:** Nick Kinney.

**Visualization:** Nick Kinney, Emma Evans.

**Writing – original draft:** Nick Kinney, Dikshya Pathak, Emma Evans, Paola Arias.

**Writing – review & editing:** Nick Kinney, Dikshya Pathak, Emma Evans, Paola Arias.

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
