## [Decision Letter · Decision Letter 0]

PONE-D-25-09713SHORT TANDEM REPEAT VARIANTS ARE ASSOCIATED WITH RNA SECONDARY STRUCTURE AND GENE EXPRESSIONPLOS ONE

Dear Dr. Kinney,

Thank you for submitting your manuscript to PLOS ONE. After careful consideration, we feel that it has merit but does not fully meet PLOS ONE’s publication criteria as it currently stands. Therefore, we invite you to submit a revised version of the manuscript that addresses the points raised during the review process. Please submit your revised manuscript by Jun 01 2025 11:59PM. If you will need more time than this to complete your revisions, please reply to this message or contact the journal office at plosone@plos.org . Please include the following items when submitting your revised manuscript:

We look forward to receiving your revised manuscript.

Kind regards,

Karthikeyan Thiyagarajan, PhD

Academic Editor

PLOS ONE

2. Please update your submission to use the PLOS LaTeX template. The template and more information on our requirements for LaTeX submissions can be found at http://journals.plos.org/plosone/s/latex .

Additional Editor Comments:

Dear Dr. Kinney,

I appreciate your work concerning the association of short tandem repeats in RNA folding and gene expression, specifically referring to the context of genetic disorders. However, there are already several studies available concerning the concept of tandem repeat-associated diseases; please check these specific articles: https://pmc.ncbi.nlm.nih.gov/articles/PMC3960014/ and https://pubmed.ncbi.nlm.nih.gov/16254211/. How is your study different from those that exist already? It is also better to discuss from a thermodynamic point of view about the fold change due to variations of STRs. Also, the intervening STRs in terminal inverted repeats and palindromes may be crucial to affect their hairpin loop kind of structure formation associated with miRNA and transposable elements. Further, you could consider the role of methylation-related specific genes, discussing the methylation and demethylation processes during the suppression and expression of the genes.

Besides, I have some additional comments and suggestions:

Have you predicted Z-DNA and H-DNA caused by short tandem repeats with your study? Because I didn't see more detail in the result and discussion.

Hydrophilic amino acids are perhaps more tolerated due to the expansion of specific STR stretches, according to Katti et al., 2001, as you have cited, so please verify whether over-represented?

Figure 1B is not clear about the cluster assignments without providing any cluster combination grouping names, from a genetic point of view, the STR-induced variations of the cluster perhaps representing alternative alleles. Please explain better and make Figure 2B self-explanatory. You could represent AA, AB, BB like this or using any specific conventions.

Have you compared the normal STR besides the fSTR with expanded stretches to the wild ones (normal STR) that affect gene expression? For instance, a similar concept in plants was explained before in plants; please see this article: https://www.nature.com/articles/ng822z#:~:text= This%20indicates%20that%2C%20in%20these,DNA%20than%20in%20repetitive%20DNA.

As you have mentioned, "fSTRs by unit are harder to interpret" because, in real time, the expression of the genes depends on many factors; however, the specific expansion of SSR due to polymerase slippage-derived errors, etc., may have a role in altered expression in exceptional cases with genetic diseases, etc. But in general, if you see even the SSR motifs from specific geographical isolates, perhaps they are more conserved than others. So, the study with fSTR with differential gene expression is perhaps more applicable to the specific individuals than others; in this case, you may need to interpret your results from Figure 3 carefully, as there are contradicting points about the accessibility.

Even if you speculate about poly AT and poly TA pairing, the folding and unfolding need specific temperature-dependent variations with specific examples, like fSTR information comparing the housekeeping genes from humans to apes, etc.

Figure 4: Is this figure derived from the similarity correlation score for the tallied pairs of fSTR alleles? You have shown the associations of different fSTR between shorter and longer variants. Is it with a CAG fSTR that more associations with gene expression occur? And how would you infer the effect due to the stability of the folding between the shorter and longer variants?

You have discussed more about the accessibility, but why didn't you consider the stability of RNA structure due to the variations of motifs as you have compared through the bpRNA-align?

You have also mentioned the use of the RNAfold program to predict thermodynamic ensembles. Why didn't you show the relevancy of the structural change with a thermodynamic point of view for specific sequences that are varying for specific fSTRs?

Large tandem repeats are also known to be associated with diseases. How do you compare them with STRs, and which major genes were focused on for differential gene expression through DESeq2?

I suggest you revise your manuscript as suggested by me and reviewers, make the contents more succinct, please try to avoid speculations, and please make the figures as self-explanatory. I also suggest comparing the species from other kingdoms; even you could compare specific housekeeping genes forms from prokaryotes to see whether there are any folding and gene expression differences due to orthologous gene comparisons, and you can also compare the specific duplicated or paralogous variations within a specific species or even with the specific genes from humans.

Reviewers' comments:

Reviewer's Responses to Questions

**Comments to the Author**

1. Is the manuscript technically sound, and do the data support the conclusions?

Reviewer #1: Partly

Reviewer #2: Yes

Reviewer #3: Yes

2. Has the statistical analysis been performed appropriately and rigorously? 

Reviewer #1: No

Reviewer #2: N/A

Reviewer #3: I Don't Know

3. Have the authors made all data underlying the findings in their manuscript fully available?

Reviewer #1: Yes

Reviewer #2: Yes

Reviewer #3: Yes

4. Is the manuscript presented in an intelligible fashion and written in standard English?

Reviewer #1: Yes

Reviewer #2: Yes

Reviewer #3: Yes

5. Review Comments to the Author

Reviewer #1: statistical analysis maybe has not been performed appropriately and rigorously, for example following:

Figure 3 – Effects of fSTR variants on RNA accessibility. Accessibility is inferred from the tally of unpaired bases

using ViennaRNA. Results of three types are obtained: (a) accessibility increases with allele length; (b) accessibility

decreases with allele length; and (c) accessibility is not associated with allele length. (d) reverse complementary

sequences tend decrease RNA accessibility.

Reviewer #2: Thank you for the opportunity to review the manuscript: "Short tandem repeat variants are associated with RNA secondary structure and gene expression". The manuscript presents a comprehensive analysis of the impact of short tandem repeat (STR) variants on RNA folding and gene expression. The study leverages data from the 1000 Genomes Project and ViennaRNA to identify transcribed STRs that affect RNA secondary structure, providing valuable insights into the role of STRs in gene regulation. The strengths of the manuscript include its robust computational approach, the large dataset used for analysis, and the novel findings regarding the association between STR variants and RNA accessibility. However, the manuscript could benefit from a more detailed discussion on the biological implications of the findings and potential experimental validation to support the computational predictions. Additionally, the characterization of STRs by motif length and gene level annotation, while thorough, could be expanded to include more diverse genomic contexts. Overall, the study makes a significant contribution to the understanding of STRs and their effects on RNA structure and gene expression, paving the way for future research in this area.

Major issues:

1. What parameters were used to identify STR variants in 2,529 samples from the 1000 genomes project? I cannot find this information in the MS. Please specify the methods and tools used for STR variant identification. Additionally, describe in detail the lengths and variations of the tandem repeats considered as STRs in this study.

2. There was any association with STRs of different repeat length?

3. Please provide all raw data including the coordinates of tested genes and STRs.

4. Would it be possible for G-quadruplexes to form in G-rich STRs?

5. Is the representation of STRs different in gene regions compared to non-gene regions? Please provide statistics on the occurrence of individual STRs.

6. The study relies heavily on computational predictions, and the lack of experimental validation is a notable weakness. Incorporating experimental data to support the computational findings would strengthen the overall conclusions and provide additional credibility to the results. If not provide, please suggest experimental design of verification in the discussion.

Reviewer #3: This manuscript presents a compelling and timely investigation into the role of short tandem repeat (STR) variants in influencing RNA secondary structure and their potential downstream effects on gene expression. The authors integrate data from the 1000 Genomes Project with RNA structure predictions using ViennaRNA, and associate RNA structural alterations with expression profiles from Geuvadis RNA-seq data. The identification of 17,446 fSTRs, including 356 with significant expression associations, contributes meaningfully to our understanding of regulatory variation beyond SNPs.

Major Comments:

1. Experimental Validation: While the computational predictions are strong, the manuscript would benefit from even a brief discussion of possible experimental strategies to validate the RNA structure changes, such as SHAPE-seq or DMS-seq.

2. Functional Implications: The discussion could be expanded to address how fSTR-induced RNA structural changes might mechanistically influence RNA processing, stability, or translation beyond splicing—particularly for those fSTRs located in UTRs or coding regions.

3. Statistical Considerations: It would be helpful if the authors described any multiple testing corrections applied during the association analysis between cluster assignments and gene expression. A false discovery rate threshold would bolster confidence in the reported 356 significant fSTRs.

4. Visualization: The figures could be improved with clearer legends and annotations, particularly for readers who may not be familiar with RNA structural clustering or exemplars.

Minor Points:

1. The manuscript would benefit from language editing in several places to improve clarity and flow, particularly in the abstract and introduction.

2. Please ensure all acronyms (e.g., fSTR, bpRNA) are defined at first use in the main text.

3. Some references for gene expression may be helpful.

a. Inhibitory Effect of Multimodal Nanoassemblies against Glycative and Oxidative Stress in Cancer and Glycation Animal Models. BioMed Research International. 2021 Apr 9;2021:8892156.

b. Revisiting global gene expression analysis. Cell. 2012 Oct 26;151(3):476-82.

6. PLOS authors have the option to publish the peer review history of their article (what does this mean? ). If published, this will include your full peer review and any attached files.

**Do you want your identity to be public for this peer review?** For information about this choice, including consent withdrawal, please see our Privacy Policy .

Reviewer #1: No

Reviewer #2: No

Reviewer #3: No

---

## [Author Response · Author response to Decision Letter 1]

28 Apr 2025

A point-by-point response to peer review comments is provided in the attached documents. We appreciate the reviewer comments and look forward to hearing back.

---

## [Decision Letter · Decision Letter 1]

PONE-D-25-09713R1SHORT TANDEM REPEAT VARIANTS ARE ASSOCIATED WITH RNA SECONDARY STRUCTURE AND GENE EXPRESSIONPLOS ONE

Dear Dr. Kinney,

Thank you for submitting your manuscript to PLOS ONE. After careful consideration, we feel that it has merit but does not fully meet PLOS ONE’s publication criteria as it currently stands. Therefore, we invite you to submit a revised version of the manuscript that addresses the points raised during the review process.

We look forward to receiving your revised manuscript.

Kind regards,

Karthikeyan Thiyagarajan, PhD

Academic Editor

PLOS ONE

Journal Requirements:

Additional Editor Comments:

Dear Authors,

Most of your revisions and responses are acceptable. The manuscript is directly based on the computational prediction of short tandem repeat-based RNA folding and its impact on gene expression, etc.

So, there is no experimental proof; however, you have added relevant experiment articles to support your analysis and hypothesis. I suggest please changing the title to "SHORT TANDEM REPEAT VARIANTS ARE POSSIBLY ASSOCIATED WITH RNA SECONDARY STRUCTURE AND GENE EXPRESSION". So it is understandable about the specific RNA folding due to tandem repeat variation for the possible associations with gene expressions with a prediction basis rather than the wet lab-related experimental approaches. Wherever required, please change throughout the manuscript to emphasize your results about the possibility of the associations with gene expressions rather than completely concluding the associations with the expression that may need additional lab experiments. One of the reviewers also suggested raw data with the coordinates of the genes tested and STRs, but your response was, "We look forward to it!" Please clarify.

Reviewers' comments:

Reviewer's Responses to Questions

**Comments to the Author**

1. If the authors have adequately addressed your comments raised in a previous round of review and you feel that this manuscript is now acceptable for publication, you may indicate that here to bypass the “Comments to the Author” section, enter your conflict of interest statement in the “Confidential to Editor” section, and submit your "Accept" recommendation.

Reviewer #1: All comments have been addressed

2. Is the manuscript technically sound, and do the data support the conclusions?

Reviewer #1: Yes

3. Has the statistical analysis been performed appropriately and rigorously? 

Reviewer #1: Yes

4. Have the authors made all data underlying the findings in their manuscript fully available?

Reviewer #1: Yes

5. Is the manuscript presented in an intelligible fashion and written in standard English?

Reviewer #1: Yes

6. Review Comments to the Author

Reviewer #1: All comments have been adequately addressed, I believe the results are meaningful. The authors tried to explain in detail.

7. PLOS authors have the option to publish the peer review history of their article (what does this mean? ). If published, this will include your full peer review and any attached files.

**Do you want your identity to be public for this peer review?** For information about this choice, including consent withdrawal, please see our Privacy Policy .

Reviewer #1: No

---

## [Author Response · Author response to Decision Letter 2]

27 May 2025

Editorial comments:

Most of your revisions and responses are acceptable. The manuscript is directly based on the computational prediction of short tandem repeat-based RNA folding and its impact on gene expression, etc. So, there is no experimental proof; however, you have added relevant experiment articles to support your analysis and hypothesis. I suggest please changing the title to "SHORT TANDEM REPEAT VARIANTS ARE POSSIBLY ASSOCIATED WITH RNA SECONDARY STRUCTURE AND GENE EXPRESSION". So it is understandable about the specific RNA folding due to tandem repeat variation for the possible associations with gene expressions with a prediction basis rather than the wet lab-related experimental approaches.

Thank you for the feedback and orchestrating peer reviews on our behalf. The title change seems very reasonable; we have updated accordingly.

Wherever required, please change throughout the manuscript to emphasize your results about the possibility of the associations with gene expressions rather than completely concluding the associations with the expression that may need additional lab experiments.

We have made several updates throughout the manuscript.

One of the reviewers also suggested raw data with the coordinates of the genes tested and STRs, but your response was, "We look forward to it!" Please clarify.

Perhaps our response was a bit too brief. The raw data used for analysis consists of several data frames (.tsv files) and python scripts (.py files). Since these items are rather large, they are more appropriate as supplementary material rather than the main text. We also look forward to sharing the code on github.com if the manuscript is published.

Reviewer #1: All comments have been adequately addressed, I believe the results are meaningful. The authors tried to explain in detail.

We appreciate the positive feedback.

---

## [Editor Report · Decision Letter 2]

SHORT TANDEM REPEAT VARIANTS ARE POSSIBLY ASSOCIATED WITH RNA SECONDARY STRUCTURE AND GENE EXPRESSION

PONE-D-25-09713R2

Dear Dr. Kinney,

We’re pleased to inform you that your manuscript has been judged scientifically suitable for publication and will be formally accepted for publication once it meets all outstanding technical requirements.

Kind regards,

Karthikeyan Thiyagarajan, PhD

Academic Editor

PLOS ONE

Additional Editor Comments:

Dear Authors,

After careful scientific evaluations with peer reviews, I am pleased to confirm the manuscript entitled "SHORT TANDEM REPEAT VARIANTS ARE POSSIBLY ASSOCIATED WITH RNA SECONDARY STRUCTURE AND GENE EXPRESSION." has been accepted for publication in PLOS ONE.

Kind regards,

Karthikeyan Thiyagarajan PhD

Academic Editor, PLOS ONE.
---

## [Editor Report · Acceptance letter]

PONE-D-25-09713R2

PLOS ONE

Dear Dr. Kinney,

I'm pleased to inform you that your manuscript has been deemed suitable for publication in PLOS ONE. Congratulations! Your manuscript is now being handed over to our production team.

Kind regards,

on behalf of

Dr. Karthikeyan Thiyagarajan

Academic Editor

PLOS ONE